# Muscle contractile exercise through a belt electrode device prevents myofiber atrophy, muscle contracture, and muscular pain in immobilized rat gastrocnemius muscle

**Yuichiro Honda**[1,2], **Ayumi Takahashi**[2], **Natsumi Tanaka**[2,3], **Yasuhiro Kajiwara**[2,4], **Ryo Sasaki**[2,5], **Seima Okita**[2,6], **Junya Sakamoto**[1,2], **Minoru Okita**[1,2]*

**1** Institute of Biomedical Sciences (Health Sciences), Nagasaki University, Nagasaki, Nagasaki, Japan, **2** Department of Physical Therapy Science, Nagasaki University Graduate School of Biomedical Sciences, Nagasaki, Nagasaki, Japan, **3** Department of Physical Therapy, School of Rehabilitation Sciences, Seirei Christopher University, Hamamatsu, Shizuoka, Japan, **4** Department of Rehabilitation, Nagasaki University Hospital, Nagasaki, Nagasaki, Japan, **5** Department of Rehabilitation, Jyuzenkai Hospital, Nagasaki, Nagasaki, Japan, **6** Department of Rehabilitation, The Japanese Red Cross Nagasaki Genbaku Hospital, Nagasaki, Nagasaki, Japan

* mokita@nagasaki-u.ac.jp

## Abstract

### Purpose

Immobilization of skeletal muscles causes muscle atrophy, muscle contracture, and muscle pain, the mechanisms of which are related to macrophage accumulation. However, muscle contractile exercise through a belt electrode device may mitigate macrophage accumulation. We hypothesized that such exercise would be effective in preventing myofiber atrophy, muscle contracture, and muscular pain. This study tested this hypothesis in immobilized rat gastrocnemius muscle.

### Materials and methods

A total of 32 rats were divided into the following control and experimental groups: immobilization (immobilized treatment only), low-frequency (LF; immobilized treatment and muscle contractile exercise with a 2 s (do) /6 s (rest) duty cycle), and high-frequency (HF; immobilized treatment and muscle contractile exercise with a 2 s (do)/2 s (rest) duty cycle). Electrical stimulation was performed at 50 Hz and 4.7 mA, and muscle contractile exercise was applied to the lower limb muscles for 15 or 20 min/session (once daily) for 2 weeks (6 times/ week). After the behavioral tests, the bilateral gastrocnemius muscles were collected for analysis.

### Results

The number of macrophages, the Atrogin-1 and MuRF-1 mRNA expression, and the hydroxyproline content in the HF group were lower than those in the immobilization and LF groups. The cross-sectional area (CSA) of type IIb myofibers in the superficial region, the PGC-1α mRNA expression, and the range of motion of dorsiflexion in the HF group were

**Data Availability Statement:** All relevant data are within the paper and its Supporting Information files.

**Funding:** No.1 ● Initials of the authors who received each award: YH ● Grant numbers awarded to each author: KAKENHI Grant No. 19K19795 ● The full name of each funder: Japan Society for the Promotion of Science ● URL of each funder website: https://www.mhlw.go.jp/stf/seisakunitsuite/bunya/hokabunya/kenkyujigyou/index.html ● Did the sponsors or funders play any role in the study design, data collection and analysis, decision to publish, or preparation of the manuscript?: The funders had no role in study design, data collection and analysis, decision to publish, or preparation of the manuscript No.2 ● Initials of the authors who received each award: MO ● Grant numbers awarded to each author: KAKENHI Grant No. 20K20668 and 21H03291 ● The full name of each funder: Japan Society for the Promotion of Science ● URL of each funder website: https://www.mhlw.go.jp/stf/seisakunitsuite/bunya/hokabunya/kenkyujigyou/index.html ● Did the sponsors or funders play any role in the study design, data collection and analysis, decision to publish, or preparation of the manuscript?: The funders had no role in study design, data collection and analysis, decision to publish, or preparation of the manuscript No.3 ● Initials of the authors who received each award: MO ● Grant numbers awarded to each author: research funding from ALCARE Co., Ltd. ● The full name of each funder: ALCARE Co., Ltd. ● URL of each funder website: https://www.alcare.co.jp/ ● Did the sponsors or funders play any role in the study design, data collection and analysis, decision to publish, or preparation of the manuscript?: The funders had no role in study design, data collection and analysis, decision to publish, or preparation of the manuscript.

**Competing interests:** Financial competing interests ● Ownership of stocks or shares: No ● Paid employment or consultancy: No ● Board membership: No ● Patent applications (pending or actual), including individual applications or those belonging to the institution to which the authors are affiliated and from which the authors may benefit: No ● Research grants (from any source, restricted or unrestricted): Yes. 1. KAKENHI from Japan Society for the Promotion of Science (grant No. 19K19795, 20K20668, and 21H03291). ◇ Description of funder's role in the study design; collection, analysis, and interpretation of data; writing of the paper; and/or decision to submit for publication: No ◇ Whether they have served or currently serve on the editorial board of the journal

significantly higher than those in the immobilization and LF groups. The pressure pain thresholds in the LF and HF groups were significantly higher than that in the immobilization group, and the nerve growth factor (NGF) content in the LF and HF groups was significantly lower than that in the immobilization group.

## Conclusion

Muscle contractile exercise through the belt electrode device may be effective in preventing immobilization-induced myofiber atrophy, muscle contracture, and muscular pain in the immobilized rat gastrocnemius muscle.

## Introduction

Immobilizing skeletal muscles causes muscle atrophy, muscle contracture, and muscle pain, which medical professionals often struggle to alleviate. We previously identified similarities in the developmental mechanisms of these muscular symptoms, in which myonuclear apoptosis produced unnecessary cytoplasm in immobilized skeletal muscles, which are phagocytosed by accumulated macrophages, resulting in muscle atrophy [1, 2]. Also, a previous study indicated that muscle protein degradation was more influential than muscle protein synthesis for muscle atrophy in 2-week immobilization [3]. Atrogin-1 and muscle RING-finger protein (MuRF)-1 as ubiquitin-proteasome pathway enzymes played the main role in muscle protein degradation [4], peroxisome proliferator-activated receptor gamma coactivator (PGC)-1α was the main regulator for Atrogin-1 and MuRF-1 [5]. Namely, the changes in factors of muscle protein degradation may affect muscle atrophy.

Additionally, accumulated macrophages enhance the expression of fibrosis-related molecules by producing interleukin (IL)-1β, which leads to muscle contracture with fibrosis via collagen overexpression [6]. Moreover, nerve growth factor (NGF), an endogenous mediator of pain, is expressed in immobilized skeletal muscles, and macrophages are the major producers of NGF [7, 8]. Therefore, macrophage accumulation is an important factor related to immobilization-induced muscle atrophy, muscle contracture, and muscle pain.

Electrical stimulation therapy has been utilized as a functional substitute for voluntary muscle contraction [9]. The usual patterns of muscle contraction by electrical stimulation are twitch and tetanic contractions, with stimulus frequencies of 1–10 Hz and 50–100 Hz, respectively, in rat skeletal muscle [10, 11]. Although a conventional electrical stimulation device often energizes the skeletal muscle through a monopolar electrode, this method has several limitations. In some situations of electrical intervention, sufficient current (power) for muscle contractile exercise may not be obtained due to the limited size of the electrodes, whereas excessive current can cause pain. Thus, an innovative electrical stimulation method with a safe and large current-carrying capacity is needed. A belt electrode-skeletal muscle electrical stimulation device (Homer Ion, Tokyo, Japan) was recently developed as a novel method to provide electrical stimulation therapy. An advantage of this device is that the belt is an electrode that can deliver electricity to the entire lower limb [12]. Thus, it is less likely to cause pain during muscle contractile exercise owing to the dispersed distribution of electricity during the intervention. Additionally, we previously demonstrated that muscle contractile exercise through a belt electrode device prevented macrophages accumulation [13]. Although muscle contractile exercise through the belt electrode device may be effective in preventing immobilization-induced muscle atrophy, contracture, and pain, this hypothesis has not been verified.

to which they are submitting: No ◇ Whether they have sat or currently sit on a committee for an organization that may benefit from publication of the paper: No 2. research funding from ALCARE Co., Ltd. ◇ Description of funder's role in the study design; collection, analysis, and interpretation of data; writing of the paper; and/or decision to submit for publication: No ◇ Whether they have served or currently serve on the editorial board of the journal to which they are submitting: No ◇ Whether they have sat or currently sit on a committee for an organization that may benefit from publication of the paper: No ● Travel grants and honoraria for speaking or participation at meetings: No ● Gifts: No Non-financial competing interests Authors do not have Non-financial competing interests. This does not alter our adherence to PLOS ONE policies on sharing data and materials.

Therefore, the present study analyzed rat gastrocnemius muscles in an experimental model to confirm this hypothesis.

## Materials and methods

### Animals

Eight-week-old male Wistar rats (CLEA Japan Inc., Tokyo, Japan) were maintained at the Center for Frontier Life Sciences of Nagasaki University. The rats were maintained in $30 \times 40 \times 20$-cm cages (two rats/cage) and exposed to a 12-h light-dark cycle at an ambient temperature of 25°C. Food and water were provided *ad libitum*. In this investigation, 32 rats ($258.9 \pm 11.5$ g) were randomly divided into an experimental group (n = 25) and a control group (n = 7). The rats in the control group were maintained without treatment or intervention. The ankle joints of the rats in the experimental group were subjected to the immobilization process described in our previous studies [7]. Briefly, the animals in the experimental group were anesthetized with the combination of 0.375 mg/kg medetomidine (Kyoritu Pharma, Tokyo, Japan), 2.0 mg/kg midazolam (Sandoz Pharma Co., Ltd., Tokyo, Japan), and 2.5 mg/kg butorphanol (Meiji Seika Pharma, Tokyo, Japan). Then, both ankle joints of each rat were fixed in full plantar flexion with plaster casts to immobilize the gastrocnemius muscle in a shortened position for 2 weeks. The plaster cast, which was fitted from above the knee joint to the distal foot, was changed weekly because of loosening owing to muscle atrophy. Additionally, the experimental groups were divided into the immobilization (n = 9; immobilized treatment only), low-frequency (LF; n = 8; immobilized treatment and muscle contractile exercise with a 2 s (do)/6 s (rest) duty cycle), and high-frequency (HF; n = 8; immobilized treatment and muscle contractile exercise with a 2 s (do)/2 s (rest) duty cycle) groups. The experimental protocol was approved by the Ethics Review Committee for Animal Experimentation at Nagasaki University (approval no. 1903281524). All experimental procedures were performed under anesthesia and efforts were made to minimize suffering.

### Protocol for electrical stimulation

According to a previous study, cyclic muscle tetanus contraction was performed using an electrical stimulator in small animals (Homer Ion, Tokyo, Japan). The electrical stimulator consisted of a control unit (for setting the stimulus cycle, frequency, and intensity) and a belt electrode. The rats in the LF and HF groups were anesthetized and the electrical stimulator was connected to a belt electrode. The belt electrodes were wrapped around the proximal thigh and distal lower leg, and the bilateral lower limb skeletal muscles were subjected to electrical stimulation with cast removal (frequency, 50 Hz). The optimum stimulus intensity and duration were determined in preliminary experiments, as outlined below.

**Preliminary experiment to determine the stimulus intensity.** The preliminary experiment included three rats. The stimulus intensity was gradually increased and plantar flexor muscle strength in the middle position of the ankle joint was measured using a force gauge. Next, 100% maximal voluntary contraction (MVC) was determined and the 60% MVC (most effective in preventing loss of muscle strength) was calculated [14]. The result of the preliminary experiment confirmed that 4.7 mA corresponded to 60% MVC (Fig 1A). Therefore, we defined the stimulus intensity as 4.7 mA.

**Preliminary experiment to determine the stimulus time.** We previously described a preliminary experiment on the 2 s (do)/6 s (rest) duty cycle [13]. Therefore, we performed a preliminary experiment with similar parameters. This preliminary experiment included three rats. The muscle strength of plantar flexion was measured (15 measurements per minute) and the stimulus time to reach <2.9 N (60% MVC) was identified. The results of this preliminary

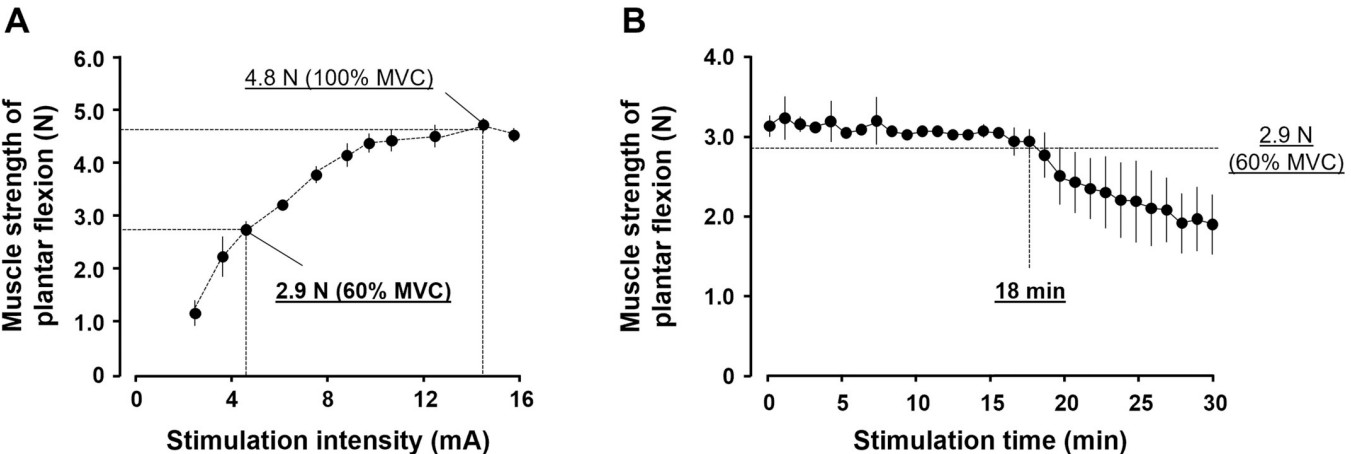

**Fig 1. Results of preliminary experiments to determine the stimulation intensity and time.** (A) Preliminary experiment to determine the stimulus intensity. The 100% and 60% maximal voluntary contraction (MVCs) were 4.8 N and 2.9 N, respectively. The results indicated that 4.7 mA corresponded to the 60% MVC. (B) Preliminary experiment to determine the stimulus time. This experiment observed the changes in the muscle strength of plantar flexion. The results showed that the strength decreased to <2.9 N at 18 min after the electrical stimulation.

experiment indicated that the muscle strength of plantar flexion decreased to <2.9 N 18 min after starting electrical stimulation (Fig 1B). Therefore, we set the stimulation time to 15 min (without muscle fatigue).

In the electrical stimulation protocol of the present study, the stimulus frequency was 50 Hz, the stimulus intensity was 4.7 mA, the duty cycle was 2 s (do)/6 s (rest) or 2 s (do)/2 s (rest), and the stimulus time was 20- or 15-min. Electrical stimulation was applied once daily for six days per week, for 2 weeks.

## Pressure pain threshold

The pressure pain threshold (PPT) of the gastrocnemius muscles was measured as the hindlimb withdrawal threshold using a Randall-Selitto apparatus equipped with a round-headed probe (tip diameter = 8 mm) [7]. The rats were tested at baseline and 1 and 2 weeks after immobilization. After cast removal, the probe was applied to the lateral head of the gastrocnemius muscle. The pressure was increased at a constant rate of 48 g/s until the animal withdrew its limb. The measurements were performed five times at intervals of at least 30 s, and the mean value, excluding the minimum and maximum values, was determined as the PPT. Immediately after this test, the plaster casts were replaced in the rats in the experimental groups.

## Range of motion of the ankle joint dorsiflexion

At 1 and 2 weeks after immobilization, the rats were anesthetized and the range of motion (ROM) of the ankle joint dorsiflexion was determined using a goniometer. The ROM was measured as the angle (0˚–180˚) between the line connecting the fifth metatarsal to the malleolus lateralis of the fibula and the line connecting the malleolus lateralis of the fibula to the center of the knee joint. The ankle was passively dorsiflexed at 0.3 N using a tension gauge (Shiro Industry, Osaka, Japan) [15].

## Tissue sampling and preparation

The left and right gastrocnemius muscles of all the rats were excised 12 hours after the last electrical stimulation. After measuring the gastrocnemius muscle wet weight, the right samples were embedded in tragacanth and the muscle sample was frozen in liquid nitrogen. Serial

frozen cross-sections of the muscles were mounted on glass slides for histological analysis. Part of the left gastrocnemius muscle was rapidly frozen in liquid nitrogen for biochemical analysis.

## Histological analysis

The cross-sections were stained with hematoxylin and eosin (H&E) (Mayer's hemalum solution, Merck KGaA, Darmstadt, Germany; Eosin Y disodium salt, Merck KGaA) and ATPase (Adenosine 5′-triphosphate disodium salt hydrate, Merck KGaA) as described previously [6, 16]. The dyed cross-sections of the muscles were then evaluated under an optical microscope. First, the H&E-stained cross-sections were used to identify myofiber morphological characteristics and signs of previous muscle injury, such as centralized nuclei. Next, the ATPase-stained cross-sectional areas (CSAs) of type I, IIa, and IIb myofibers were analyzed using Scion Image software (National Institutes of Health, MD, USA). More than 100 myofiber measurements were recorded per animal [6, 17].

## Immunohistochemical analysis

The cross-sections were air-dried and fixed in ice-cold acetone for 10 min. To inhibit endogenous peroxidase activity, the sections were incubated with 0.3% $H_2O_2$ in methanol for 40 min at 37˚C. After washing with 0.01 M phosphate-buffered saline (PBS pH 7.4), the sections were incubated for 10 min at 37˚C with 0.1% Triton X-100 in PBS. The sections were then blocked with 5% bovine serum albumin in PBS for 60 min and incubated overnight at 4˚C with mouse anti-CD11b primary antibody (1:2000; BMA Biomedicals, Augst, Switzerland). The sections were rinsed in PBS for 15 min and incubated with biotinylated goat anti-mouse IgG (1:1000; Vector Laboratories) for 60 min at 37˚C. The sections were then rinsed with PBS and allowed to react with avidin-biotin-peroxidase complexes (VECTASTAIN Elite ABC kit, Vector Laboratories) for 60 min at 37˚C. Horseradish peroxidase binding sites were visualized with 0.05% 3,3-diaminobenzidine and 0.01% $H_2O_2$ in 0.05 M Tris buffer at 37˚C. After the final washing step, the CD11b sections were stained with eosin and observed under an optical microscope. Using microscopy and standardized light conditions, the sections were magnified 400× (CD11b) and images were captured with a digital camera (Nikon, Tokyo, Japan). The number of macrophages was determined from the 400× images by counting the number of CD11b-positive cells per 100 muscle fibers. Vascular areas were excluded from the analysis [6, 13].

## Molecular biological analysis

The soleus muscles were used for this analysis. Total RNA was extracted from muscle samples using a RNeasy Fibrous Tissue Mini Kit (Qiagen, CA, USA). Total RNA was used as a template with a QuantiTect® Reverse Transcription Kit (Qiagen) to prepare cDNA, and real-time RT-PCR was performed using Brilliant III Ultra-Fast SYBR Green QPCR Master Mix (Agilent Technologies, CA, USA). The cDNA concentration of all samples was unified to 25 ng/μl, the cDNA was applied 0.2 μl to each well. The synthetic gene-specific primers are listed in Table 1. The threshold cycle (Ct) was determined using an Mx3005P Real-Time QPCR System (Agilent Technologies). The mRNA expression of target genes was calculated using the ΔΔct method.

## Biochemical analysis

The gastrocnemius muscles were assessed for hydroxyproline expression (a parameter for collagen expression) using our previous method [14]. Briefly, the muscle samples were immersed in 1.0 M PBS (pH 7.4) and homogenized using a Micro Smash™ device (MS-100R; Tomy, Tokyo, Japan). Subsequently, the muscle samples were hydrolyzed in 6 N HCl for 15 h at

**Table 1. Arrangement of synthetic gene-specific primers.**

| Object gene | Arrangement | | Gene Bank No. |
|---|---|---|---|
| | Forward | Reverse | |
| PGC-1α | 5'- CAAGCCAAACCAACAACTTTATCTCT -3' | 5'- CACACTTAAGGTTCGCTCAATAGT -3' | NC051349.1 |
| Atrogin-1 | 5'-ACTAAGGAGCGCCATGGATACT-3' | 5'-GTTGAATCTTCTGGAATCCAGGAT-3' | AY059628.1 |
| MuRF-1 | 5'-TGACCAAGGAAAACAGCCACCAG-3' | 5'-TCACTCTTCTTCTCGTCCAGGATGG-3' | AY059627.1 |
| β-actin | 5'-GTGCTATGTTGCCCTAGACTTCG-3' | 5'-GATGCCACAGGATTCCATACCC-3' | BC063166.1 |

PGC, peroxisome proliferator-activated receptor gamma coactivator; MuRF, muscle RING-finger protein

110°C and then dried in 6 N HCl using an evaporator (EZ-2 HCL-resistant model; Ikeda Scientific, Tokyo, Japan). The muscle samples were hydrolyzed in NaOH for 1 h at 90°C. The hydrolyzed specimens were then mixed with buffered chloramine-T reagent and subsequently oxidized at 20°C. The chromophore was developed by adding Ehrlich's aldehyde reagent. The absorbance of each sample was measured at 540 nm using a SpectraMax 190 spectrophotometer (Molecular Devices, CA, USA). The absorbance values were plotted against the concentration of the hydroxyproline standard. The concentrations of hydroxyproline in the unknown sample extracts were determined using a standard curve and calculated as content per dry weight (μg/mg dry weight).

## Enzyme-linked immunosorbent assay (ELISA)

The gastrocnemius muscle was homogenized in cold lysis buffer (pH 8.0) at 4°C. The homogenate was centrifuged for 20 min at 12,000 rpm at 4°C, and the supernatant was stored at −80°C. NGF protein expression in the gastrocnemius muscle was examined using an ELISA kit (Boster Biological Technology, Pleasanton, California, USA) with a range of validity of 15.6–1,000 pg/mL. The protein content of each tissue supernatant was determined using a bicinchoninic acid protein assay kit (Pierce, Rockford, Illinois, USA). The NGF levels were normalized to the protein content.

## Statistical analysis

All data are presented as means ± standard deviation. The ROM of dorsiflexion and PPT were were assessed using two-way ANOVA, followed by Scheffé's method. Also, the differences between groups of other parameters were assessed using one-way ANOVA, followed by Scheffé's method. Differences were considered statistically significant at $p < 0.05$.

## Results

### Macrophages

The control group showed 5.8 ± 1.0 CD11b-positive cells per 100 myofibers at 2 weeks. In the immobilization, LF, and HF groups, the values were 18.9 ± 0.9, 18.3 ± 1.0, and 14.6 ± 0.8, respectively, at 2 weeks after immobilization (Fig 2A and 2B). The numbers of CD11b-positive cells in the experimental groups were significantly higher than that in the control group and lower in the HF group than in the immobilization and LF groups.

### Body weight, muscle wet weight, and relative weight ratio

The body weight, muscle wet weight, and relative weight ratio (muscle wet weight per body weight) at 2 weeks are shown in Table 2. All parameters were significantly lower in the

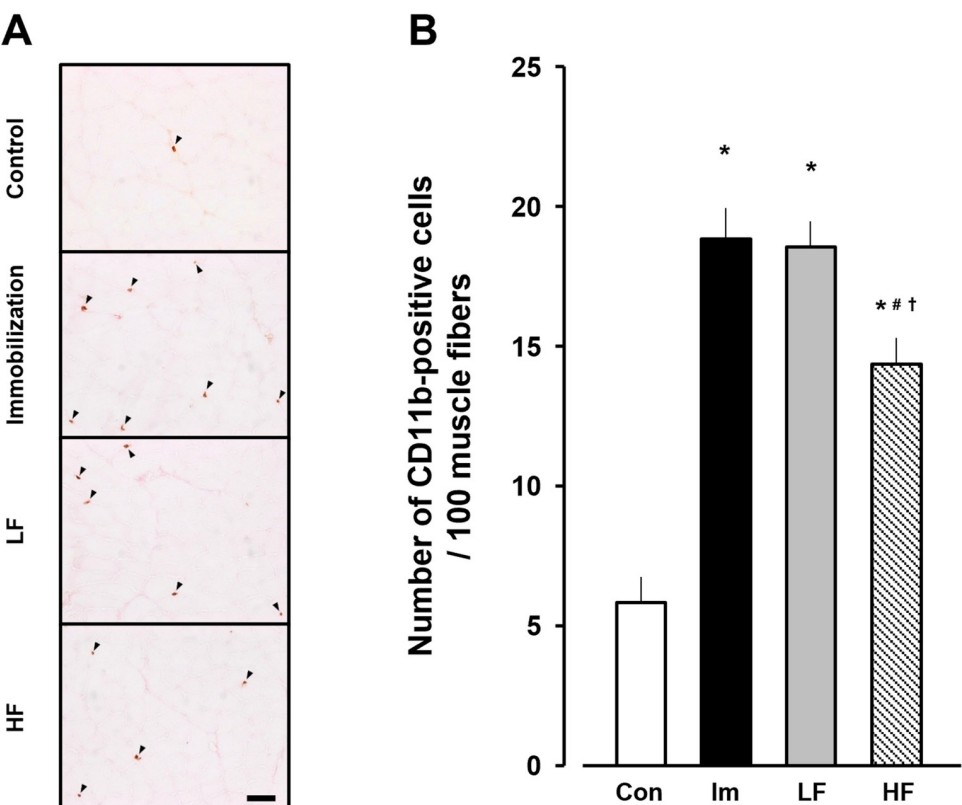

**Fig 2. Numbers of CD11b-positive cells in gastrocnemius muscles.** (A) Immunohistochemical staining images of CD11b in gastrocnemius muscles. Arrowheads, CD11b-positive cells. Scale bar, 50 μm. (B) The number of CD11b-positive cells per 100 myofibers. Open bar, control group (Con). Black bar, immobilization group (Im). Gray bar, LF group. Hatched bar, HF group. Data are presented as mean ± standard deviation. *Significant difference (p < 0.05) compared to the control group #Significant difference (p < 0.05) compared to the immobilization group. †Significant difference (p < 0.05) compared to the LF group. LF, low-frequency; HF, high-frequency.

experimental groups than those in the control group but did not differ significantly among the immobilization, LF, and HF groups.

## H&E imaging and CSA

The H&E-stained cross-sections of the experimental groups showed no abnormal findings, except for atrophic changes (Fig 3). Assessment of the CSAs in the deep and superficial regions of type I, IIa, and IIb myofibers (Fig 4A) showed a CSA of type I myofibers was 2503.3 ± 491.0 μm$^2$ in the deep region of ATPase-stained cross-sections in the control group. The CSAs of type I myofiber in the immobilization, LF, and HF groups were 1113.6 ± 144.6,

**Table 2. Body weight, muscle wet weight, and relative weight ratio at 2 weeks.**

|  | Control | Immobilization | LF | HF |
|---|---|---|---|---|
| BW (g) | 296.7 ± 11.5 | 245.8 ± 13.5* | 235.0 ± 13.7* | 233.8 ± 7.7* |
| MWW (mg) | 663.2 ± 80.5 | 407.2 ± 47.4* | 418.0 ± 56.4* | 407.4 ± 31.2* |
| RWR (mg/g) | 2.2 ± 0.2 | 1.7 ± 0.1* | 1.8 ± 0.2* | 1.7 ± 0.1* |

BW: body weight, MWW: muscle wet weight, RWR: relative weight ratio

* p < 0.05 compared to the control group.

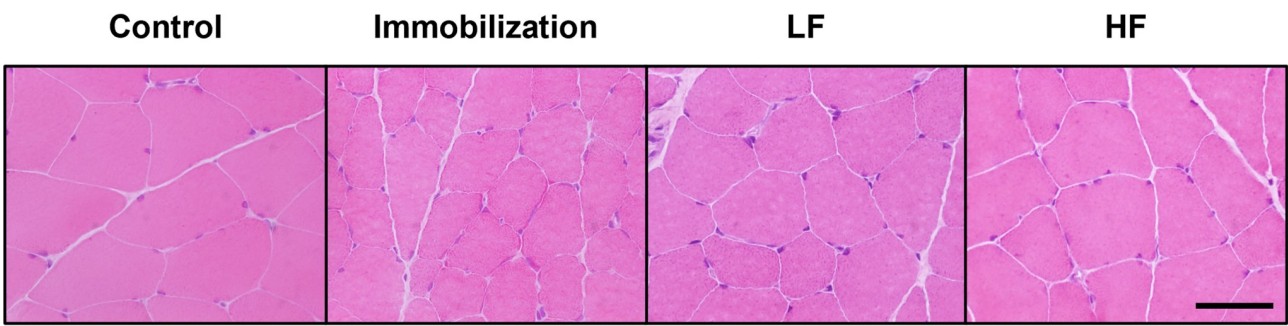

**Fig 3. Hematoxylin and eosin-stained imaging of rat gastrocnemius muscles.** No abnormal findings are visible in the experimental groups, except for atrophic changes. Scale bar, 50 μm.

1176.0 ± 126.7, and 1351.8 ± 147.5 μm², respectively, at 2 weeks after immobilization (Fig 4B). The CSA of type IIa myofibers in the deep region was 1783.1 ± 387.4 μm² in the control group. The CSAs of type IIa myofibers in the immobilization, LF, and HF groups were 934.1 ± 138.2, 938.8 ± 112.8, and 1020.5 ± 129.2 μm², respectively, at 2 weeks after immobilization (Fig 4C). The CSA of type IIb myofibers of the deep region was 2344.9 ± 572.5 μm² in the control group. The CSAs of type IIb myofiber in the immobilization, LF, and HF groups were 1275.7 ± 103.3, 1266.7 ± 169.9, and 1477.4 ± 118.6 μm², respectively, at 2 weeks after immobilization (Fig 4D).

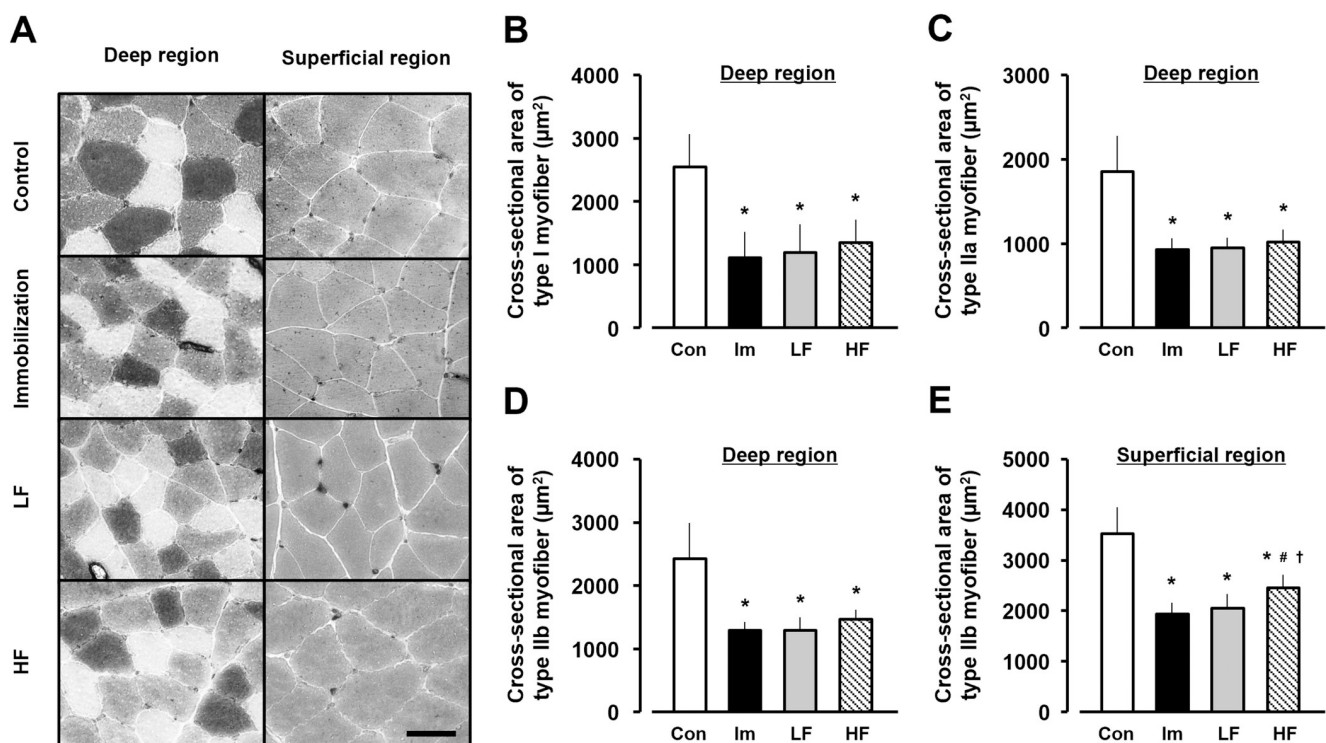

**Fig 4. Cross-sectional areas (CSAs) in the deep and superficial regions of gastrocnemius muscles.** (A) ATPase staining of gastrocnemius muscle. Black areas, type I fibers. White areas, type IIa fibers. Gray areas, type IIa fibers. Scale bar, 50 μm. (B) CSA of type I fibers in the deep region. (C) CSA of type IIa fibers in the deep region. (D) CSA of type IIb fibers in the deep region. (E) CSA of type IIb fibers in the superficial region. Open bars, control group (Con). Black bars, immobilization group (Im). Gray bars, LF group. Hatched bars, HF group. Data are presented as mean ± standard deviation. *Significant difference (p < 0.05) compared to the control group #Significant difference (p < 0.05) compared to the immobilization group. †Significant difference (p < 0.05) compared to the LF group. LF, low-frequency; HF, high-frequency.

In the superficial region of ATPase-stained cross-sections, the CSA of type IIb myofibers was $3516.3 \pm 536.2 \ \mu m^2$ in the control group. The CSAs of type IIb myofiber in the immobilization, LF, and HF groups were $1920.4 \pm 201.9$, $1969.5 \pm 253.5$, and $2439.8 \pm 253.0 \ \mu m^2$, respectively, at 2 weeks after immobilization (Fig 4E).

In the deep region of the gastrocnemius muscle, the CSAs of type I, IIa, and IIb myofibers in the experimental groups were significantly lower than those in the control group and did not differ significantly among the immobilization, LF, and HF groups. In addition, in the superficial region of the gastrocnemius muscle, the CSAs of type IIb myofibers in the experimental groups were significantly lower than that in the control group, whereas the CSA of type IIb myofibers in the HF group was significantly higher than those in the immobilization and LF groups.

## PGC-1α, Atrogin-1, MuRF-1 mRNA expression

The PGC-1α mRNA expression was $1.1 \pm 0.2$ in the control group. In the immobilization, LF, and HF groups, the expression was $0.6 \pm 0.1$, $0.7 \pm 0.1$, and $1.0 \pm 0.1$ respectively, at 2 weeks after immobilization (Fig 5A). The Atrogin-1 mRNA expression was $1.1 \pm 0.4$ in the control group. In the immobilization, LF, and HF groups, the expression was $2.5 \pm 0.4$, $2.2 \pm 0.4$, and $1.7 \pm 0.4$ respectively, at 2 weeks after immobilization (Fig 5B). The MuRF-1 mRNA expression was $1.1 \pm 0.1$ in the control group. In the immobilization, LF, and HF groups, the expression was $2.2 \pm 0.5$, $2.0 \pm 0.4$, and $1.5 \pm 0.3$ respectively, at 2 weeks after immobilization (Fig 5C).

The PGC-1α mRNA expression in the immobilization and LF groups was significantly lower than that in the control group, that in the control and HF groups was no significant difference. Additionally, the Atrogin-1 and MuRF-1 mRNA expressions in the experimental groups were significantly higher than those in the control group, whereas those in the HF group were significantly lower than those in the immobilization and LF groups.

## ROM of ankle joint dorsiflexion

The ROM of dorsiflexion in all groups was 160˚ at baseline, and was 160˚ in the control group during both experimental periods. At 1 and 2 weeks after immobilization, the ROM values

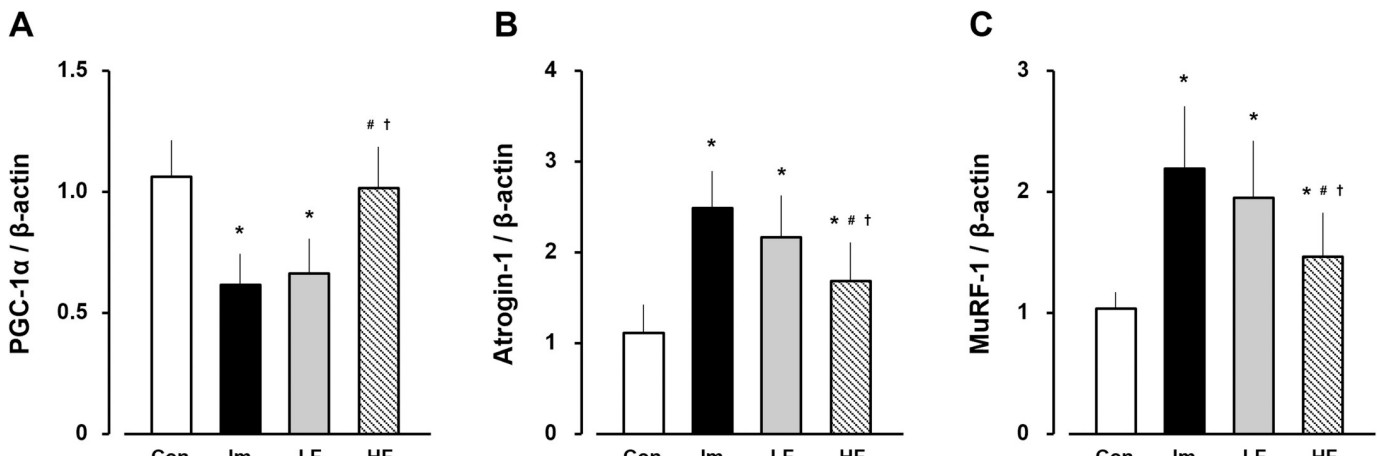

**Fig 5.** mRNA expression of PGC-1α (A), Atrogin-1 (B), and MuRF-1 (C) in gastrocnemius muscle. Open bars, control group (control). Black bars, immobilization group (Im). Gray bars, LF group. Hatched bars, HF group. Data are presented as mean ± standard deviation. *Significant difference (p < 0.05) compared to the control group #Significant difference (p < 0.05) compared to the immobilization group. †Significant difference (p < 0.05) compared to the LF group. LF, low-frequency; HF, high-frequency.

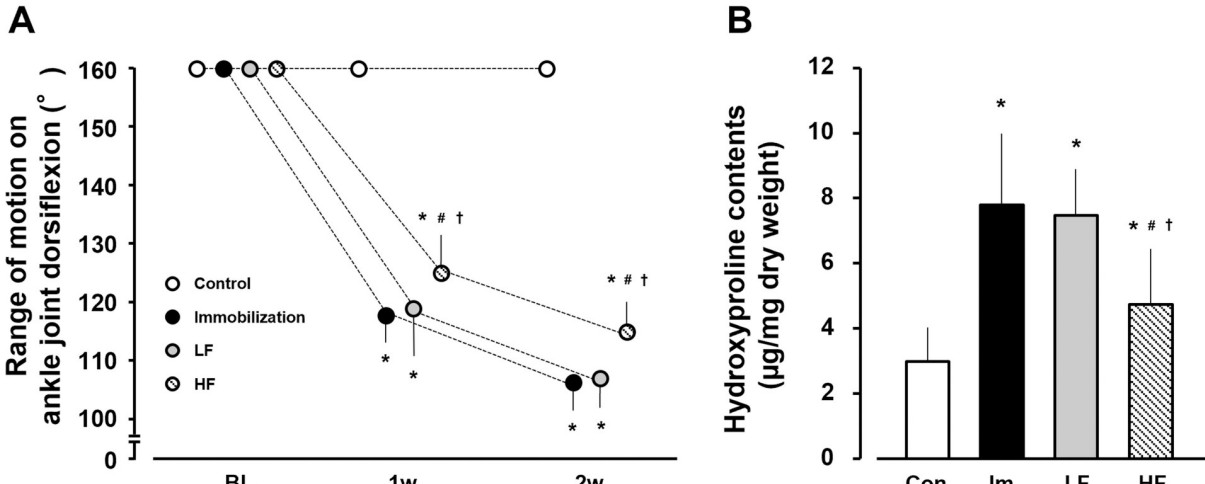

**Fig 6.** Range of motion of the ankle joint on dorsiflexion (A) and hydroxyproline content (B) in gastrocnemius muscle. Open circles and bars, control group (control). Black circles and bars, immobilization group (Im). Gray circles and bars, LF group. Hatched circles and bars, HF group. Data are presented as mean ± standard deviation. *Significant difference (p < 0.05) compared to the control group #Significant difference (p < 0.05) compared to the immobilization group. †Significant difference (p < 0.05) compared to the LF group. LF, low-frequency; HF, high-frequency.

were 118.1 ± 4.6˚ and 106.4˚± 4.8˚, 119.1 ± 8.0˚ and 107.2 ± 4.8˚, and 124.4˚± 6.0˚ and 115.9 ± 4.2˚ in the immobilization, LF, and HF groups, respectively (Fig 6A). During the 1- and 2-week experimental periods, the ROMs of dorsiflexion in the experimental groups were significantly lower than that in the control group and was higher in the HF group than in the immobilization and LF groups.

## Hydroxyproline content

The hydroxyproline content was 3.2 ± 0.9 μg/mg dry weight in the control group and 7.8 ± 2.1, 7.4 ± 1.4, and 4.9 ± 1.7 μg/mg dry weight, respectively, in the immobilization, LF, and HF groups at 2 weeks after immobilization (Fig 6B). The hydroxyproline content in the experimental groups was significantly higher than that in the control group and lower in the HF group than in the immobilization and LF groups.

## PPT

The PPT of the gastrocnemius muscles in the control group was 210.2 ± 9.6 g at baseline, 221.8 ± 9.5 g at 1 week, and 230.9 ± 7.3 g at 2 weeks. In the immobilization group, the PPT was 207.1 ± 8.2 g at baseline, 167.6 ± 16.0 g at 1 week, and 161.0 ± 7.2 g at 2 weeks. In the LF group, the PPT was 208.2 ± 17.5 g at baseline, 188.4 ± 7.7 g at 1 week, and 194.9 ± 4.7 g at 2 weeks. In the HF group, the PPT was 205.8 ± 13.6 g at baseline, 188.4 ± 9.3 g at 1 week, and 191.8 ± 5.0 g at 2 weeks (Fig 7A). During the 1- and 2-week experimental periods, the PPT values of the gastrocnemius muscles in the experimental groups were significantly lower than that in the control group and were higher in the LF and HF groups than in the immobilization group.

## NGF content

The NGF content was 9.6 ± 2.3 pg/mg dry weight in the control group at 2 weeks. In the immobilization, LF, HF groups, the content was 30.2 ± 5.0, 19.7 ± 3.6, and 21.0 ± 2.9 pg/mg dry weight, respectively, at 2 weeks after immobilization (Fig 7B). The hydroxyproline content

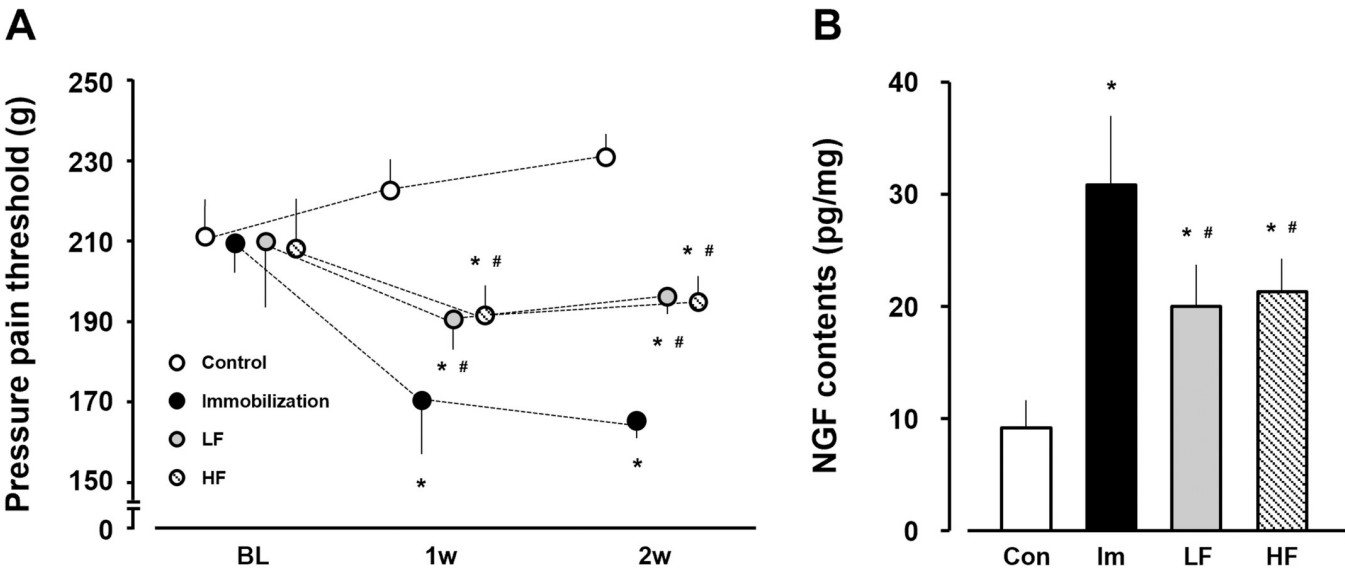

**Fig 7.** Pressure pain threshold (A) and NGF content (B) in gastrocnemius muscle. Open circles and bars, control group (control). Black circles and bars, immobilization group (Im). Gray circles and bars, LF group. Hatched circles and bars, HF group. Data are presented as mean ± standard deviation. *Significant difference (p < 0.05) compared to the control group #Significant difference (p < 0.05) compared to the immobilization group. NGF, nerve growth factor; LF, low-frequency; HF, high-frequency.

was significantly lower in the HF group than those in the immobilization and LF groups, and did not differ significantly between the HF and control groups.

## Discussion

This study investigated the biological effects of muscle contractile exercise using a belt electrode device to prevent immobilization-induced myofiber atrophy, muscle contracture, and muscular pain based on behavioral, histological, biochemical, and immunohistochemical analyses.

In the immobilization group, the number of CD11b-positive cells increased 3.3-fold compared to that in the control group at 2 weeks after immobilization, consistent with the results of previous studies [7]. The mechanism underlying the increase in the number of macrophages has been described previously. Monocyte chemoattractant protein-1 (MCP-1) plays a key role in monocyte/macrophage migration [18]. In previous studies, MCP-1 mRNA expression in the gastrocnemius muscle was higher on the immobilized side than on the control side after 2 weeks of immobilization [19]. Additionally, in the soleus muscles of the same immobilized model, MCP-1 mRNA expression and the number of CD11b-positive cells increased after 2 weeks of immobilization. Our findings and those of previous studies regarding CD11b-positive cells demonstrated macrophage accumulation in immobilized gastrocnemius muscles. The relative weight ratio and CSA of the deep and superficial regions in the immobilization group were significantly lower than that in the control group. Previous reports demonstrated a decreased relative weight ratio in immobilized rat gastrocnemius muscles [20]. In addition, previous analysis of the CSAs of both regions showed significantly lower type I, IIa, and IIb myofiber diameters in immobilized gastrocnemius muscles compared to the control [10, 21]. Moreover, the PGC-1α mRNA expression in the immobilization group was significantly lower than that in the control group, the Atrogin-1 and MuRF-1 mRNA expression in the immobilization group were significantly higher than those in the control group. The downregulation of PGC-1α led to the overexpression of Atrogin-1 and MuRF-1, these alterations induced muscle

fiber atrophy [5]. Furthermore, macrophage accumulation in skeletal muscle plays an important role in mediating the development of muscle fiber atrophy [22]. Therefore, we surmised that macrophage accumulation and changes in factors of muscle protein degradation were associated with immobilization-induced myofiber atrophy.

Hydroxyproline is a unique amino acid composed of collagen; thus, increased hydroxyproline content indicates collagen overexpression known as fibrosis [23]. Reduction in muscle extensibility decreases joint mobility, thus contributing to muscle contracture. Fibrosis in skeletal muscle is strongly associated with reduced muscle extensibility [15, 24]. In addition, myofibroblasts produce large amounts of collagen and play a major role in pathological contracture. The upregulation of fibrosis-related factors via macrophage accumulation induces fibroblast differentiation into myofibroblasts [25, 26]. The results of the present study revealed significantly higher hydroxyproline content in the immobilization group than that in the control group, as well as a significantly lower ROM of dorsiflexion in the immobilization group than that in the control group. Fibrosis via macrophage accumulation may be related to the development of muscle contracture in immobilized gastrocnemius muscles.

NGF is a strong pain mediator, with macrophages the main producing cells. NGF expression and hyperalgesia are associated with pathological muscle pain [27, 28]. Additionally, intramuscular injection of NGF decreased PPT in a dose-dependent manner in skeletal muscles [27, 29]. Furthermore, our findings suggested that the injection of an NGF receptor inhibitor increased PPT levels in immobilized gastrocnemius muscles [7]. Thus, NGF upregulation along with macrophage accumulation may be involved in immobilization-induced muscle pain.

The HF treatment more effectively prevented muscle atrophy and contracture than LF, while both treatment groups showed similar preventive effects on muscle pain. The number of CD11b-positive cells in the HF group was significantly lower than those in the immobilization and LF groups. A previous report showed that physical exercise reduced macrophage response in the myocardium [30]. Additionally, electrical stimulation of muscle contractile exercise suppressed MCP-1 upregulation and macrophage accumulation [10]. Moreover, >200 muscle contractile exercises via electrical stimulation were required per set to maintain CSA; this alteration was caused by preventing macrophage accumulation [13, 31]. In the present study, the LF and HF groups performed 150 and 225 muscle contractile exercises per set, respectively. Therefore, sufficient muscle contractile exercise via electrical stimulation was applied to the immobilized gastrocnemius muscle only in the HF group, and this treatment may have suppressed macrophage accumulation. Macrophages phagocytose unnecessary cytoplasm via myonuclear apoptosis, which induces muscle fiber atrophy [1, 2]. Mechanical stimuli via electrical stimulation inhibit apoptosis by activating several signaling pathways [32]. Electrical stimulation prevents cell apoptosis by regulating pro- and anti-apoptotic proteins [33, 34]. Furthermore, muscle contractile exercise by electrical stimulation suppresses the reduction of myonuclei, and macrophage accumulation is mitigated in immobilized skeletal muscles [13]. In short, muscle contractile exercise with electrical stimulation may prevent a decrease in myonuclei via apoptosis, which may lead to the suppression of muscle fiber atrophy caused by macrophage accumulation. On the other hand, the downregulation of PGC-1α and the overexpression of Atrogin-1 and MuRF-1 were suppressed in the HF group. Muscle contractile exercise was necessary to keep PGC-1α homeostasis, this led to prevent the upregulation of ubiquitin-proteasome pathway enzymes [35]. Therefore, we surmised that the same alterations as above occurred in HF group. However, these beneficial effects were confirmed only in the superficial region of the gastrocnemius muscle. The motor unit of type IIb myofibers is larger than that of type I and IIa myofibers, and the electrical resistance is small in skeletal muscles with large motor units [36]. The unique effect of electrical stimulation on skeletal muscle is the

reversal of the recruitment pattern, which is typically associated with voluntary muscle activation. Muscle contractile exercise with electrical stimulation occurs strongly in type IIb myofibers on the superficial region of skeletal muscles [37, 38]. Based on these reports, we surmised that muscle contractile exercise with electrical stimulation was effective in preventing type IIb myofiber atrophy in the superficial region of immobilized rat gastrocnemius muscles.

Macrophage accumulation affected the upregulation of fibrosis-related factors, such as IL-1β and transforming growth factor (TGF)-β1, etc. [6]. However, in immobilized skeletal muscles subjected to muscle contractile exercise, the upregulation of IL-1β and TGF-β1 due to macrophage accumulation is suppressed [13]. Thus, the differentiation of fibroblasts into myofibroblasts via IL-1β/TGF-β1 signaling is disturbed and fibrosis, which is the main lesion of immobilization-induced muscle contracture, is prevented [13]. In the present study, the hydroxyproline content in the HF group was significantly lower than those in the immobilization and LF groups, and the ROM of dorsiflexion in the HF group was the highest among the experimental groups. Therefore, muscle contractile exercises based on the HF protocol may be effective in preventing immobilization-induced muscle contracture.

The NGF content in the HF group was significantly lower than that in the immobilization group. Macrophage accumulation is mitigated in immobilized skeletal muscles [13, 31]. In addition, the reduction of macrophages induces NGF downregulation [39] and decreases NGF expression related to increased PPT in skeletal muscles [27]. The results of this study revealed higher PPT in the HF group than that in the immobilization group. Muscle contractile exercise with the HF protocol may prevent NGF upregulation via macrophage accumulation; moreover, these alterations may be involved in the suppression of immobilization-induced muscle pain.

This study had several limitations. First, it is uncertain whether the current electrical stimulation protocol is the most effective. Further examination of various frequencies, intensities, duty cycles, times, and intraday sessions of electrical stimulation protocols is required. Additionally, this study was unable to determine why contractile exercise through the LF protocol showed a beneficial effect on the PPT of the gastrocnemius muscles. Further studies on the detailed polarization changes in M1 and M2 macrophages in the gastrocnemius muscles of the LF and HF groups are required to address this issue. Moreover, the present study could not confirm whether myonuclear apoptosis, a key lesion in macrophage accumulation, was suppressed by the muscle contractile exercise in the HF protocol. Future studies using terminal deoxynucleotidyl transferase dUTP nick end labeling (TUNEL) staining of muscle sections are required to answer this question. Finally, data related to the causal relationship between cellular and molecular events were insufficient. Therefore, future studies using antagonists or inhibitors are warranted to address this limitation.

In summary, muscle contractile exercise through a belt electrode device may be effective in preventing immobilization-induced myofiber atrophy, muscle contracture, and muscular pain in immobilized rat gastrocnemius muscles.

## Supporting information

**S1 Data.**
(XLSX)

## Acknowledgments

The belt electrode-skeletal muscle electrical stimulation system was provided by Homer Ion Co. Ltd. (Tokyo, Japan). We would like to thank Editage (www.editage.com) for English language editing.

## Author Contributions

**Conceptualization:** Yuichiro Honda, Junya Sakamoto, Minoru Okita.

**Data curation:** Yuichiro Honda, Junya Sakamoto, Minoru Okita.

**Formal analysis:** Yuichiro Honda.

**Funding acquisition:** Yuichiro Honda.

**Investigation:** Yuichiro Honda, Ayumi Takahashi, Natsumi Tanaka, Yasuhiro Kajiwara, Ryo Sasaki, Seima Okita.

**Methodology:** Yuichiro Honda, Ayumi Takahashi, Natsumi Tanaka, Yasuhiro Kajiwara, Ryo Sasaki, Seima Okita.

**Project administration:** Yuichiro Honda, Minoru Okita.

**Resources:** Minoru Okita.

**Software:** Minoru Okita.

**Supervision:** Yuichiro Honda, Junya Sakamoto, Minoru Okita.

**Validation:** Yuichiro Honda, Ayumi Takahashi, Natsumi Tanaka, Yasuhiro Kajiwara, Ryo Sasaki, Seima Okita, Junya Sakamoto, Minoru Okita.

**Visualization:** Yuichiro Honda.

**Writing – original draft:** Yuichiro Honda.

**Writing – review & editing:** Yuichiro Honda.

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
