## [Decision Letter · Decision Letter 0]

25 Jun 2022

PONE-D-22-09461Muscle contractile exercise through a belt electrode device prevents myofiber atrophy, muscle contracture, and muscular pain in immobilized rat gastrocnemius musclePLOS ONE

Dear Dr. Honda,

Thank you for submitting your manuscript to PLOS ONE. After careful consideration, we feel that it has merit but does not fully meet PLOS ONE’s publication criteria as it currently stands. Therefore, we invite you to submit a revised version of the manuscript that addresses the points raised during the review process.

You will see that the reviewers have raised a range of concerns (please further below), from minor aspects on formatting to more substantial aspects on methodological clarity and approaches, which could alter the results and their interpretation. Therefore, I would invite you to carefully consider each comment and address it appropriately.  

We look forward to receiving your revised manuscript.

Kind regards,

Theodoros M. Bampouras

Academic Editor

PLOS ONE

Journal Requirements: 

**Comments to the Author**

1. Is the manuscript technically sound, and do the data support the conclusions?

Reviewer #1: Yes

Reviewer #2: Yes

2. Has the statistical analysis been performed appropriately and rigorously? 

Reviewer #1: Yes

Reviewer #2: No

3. Have the authors made all data underlying the findings in their manuscript fully available?

Reviewer #1: Yes

Reviewer #2: No

4. Is the manuscript presented in an intelligible fashion and written in standard English?

Reviewer #1: Yes

Reviewer #2: Yes

5. Review Comments to the Author

Reviewer #1: Dear Editor,

Re: "Muscle contractile exercise through a belt electrode device prevents myofiber atrophy, muscle contracture, and muscular pain in immobilized rat gastrocnemius muscle "

Thank you for my opportunity to review this manuscript. The study used the immobilized rat gastrocnemius muscle model to show the consequences of the belt electrode device on skeletal muscle. The device was used as the therapeutic method to alleviate the myofiber atrophy, muscle contracture, and muscular pain effects caused by immobilization. The objectives of the study are clearly stated, the concept and the experimental design of this study is well structured and the experiments are well conducted. However, there are points need to be addressed to increase the reliability and quality of the manuscript.

Reviewer Comments:

1. The name of the abscissa axis in Figure 1. B is wrong, and the author wants to express the stimulation time, not the stimulation intensity.

2. The special symbols for p-values denoting between-group differences in Figure 2. B and Figure 5. B do not exactly match the legends.

3. According to the author's experimental results in Figure 4. A, we found that there are only type IIb muscle fibers in the superficial gastrocnemius muscle in this experiment. How to determine that there are only type IIb muscle fibers in the superficial gastrocnemius muscle?

4. In the text, the baseline PPT of the gastrocnemius muscles in the LF group was higher than that in the immobilization group. However, in Figure 6. A, the gastrocnemius muscle baseline PPT value in the LF group was lower than in the fixed group.

Reviewer #2: This study was to test the physiological benefit of repeated muscle contractile exercise using a belt electrode device in an animal model with both ankle joints immobilization. The authors reported that myofiber atrophy, joint movement limitation, and mechanical muscle hyperalgesia were observed in immobilized rats, and muscle contractile exercise through the belt electrode device was beneficiary in preventing them. In general, the study has translational significance followed with previous studies. The method was detailed, and data were presented clearly to support the conclusion. However, there are some major comments to improve this paper before I would recommend that the paper be published. If the authors can address my concerns, I am happy to review this paper again. In addition, here are some minor essential changes needed and some suggestions to consider.

Major essential changes needed:

1. In this study, myonuclear apoptosis is mentioned as one of the mechanisms of skeletal muscle atrophy, but the involvement of muscle protein synthesis and degradation is ignored. The total and phosphorylated forms of Akt, p70S6K, and 4E-BP1 (FoxO 1/3 phosphorylation would also be informative) in each experimental group should be added as items to be examined. For example, the time course of the acute effect of a single bout of muscle contractile exercise through the belt electrode device on their phosphorylation should be investigated. In particular, Akt-mTOR signaling pathways may orchestrate both a positive protein metabolism in skeletal muscle and myonuclear apoptosis.

2. Methods. In statistical analysis for "range of motion on ankle joint dorsiflexion" and "pressure pain threshold", a split-plot ANOVA using 2 factors (time [baseline vs. 1W vs. 2W] and group [control vs. immobilization vs. LF vs. HF]) must be used to determine the interaction and main effects, after verifying the normality of the data.

Minor essential changes needed:

1. Methods. Please add the length of time between last electrical stimulation, and euthanasia and tissue collection. Exercise itself can raise macrophage accumulation, NGF upregulation, and some cytokines in tissues. This will help exercise physiology readers interpret the results compared to their own.

2. Methods. The authors described "2 sec (do)/6 sec (rest) duty cycle" as low-frequency and "2 sec (do)/2 sec (rest) duty cycle" as high-frequency for electrical stimulation conditions. It is necessary to clarify the definition of low-frequency or high-frequency in the electrical stimulation conditions.

3. Methods. To mention that 60% maximal voluntary contraction is most effective in preventing skeletal muscle atrophy, should be provided a clear link to the previous studies referenced. In general, it is defined not only by the intensity of muscle contraction exercise, but also by the number of exercises and the number of sets.

4. Methods. What is the stimulus intensity you set when you conducted the preliminary experiment to determine the stimulus time?

5. Results. For histological observation, the quality of the tissue images stained with immunohistochemistry (Figure 2A) and H&E (Figure 3) were poor in the PDF file.

Minor suggested changes:

1. Introduction. Hypotheses should be added at the end of the Introductory section.

2. Results. In lines 224 on page 10 to line 226 on page 11, "In the immobilization, LF, and HF groups..." seems to be more appropriate than "In the immobilization, LF, HF groups...".

3. Results. In lines 253-254 on page 12, “The CSA of type IIa myofibers was...” could be more accurate if change to “The CSA of type IIa myofibers in the deep region was...”.

6. PLOS authors have the option to publish the peer review history of their article (what does this mean?). If published, this will include your full peer review and any attached files.

Reviewer #1: **Yes: **Yun Zhou

Reviewer #2: No

---

## [Author Response · Author response to Decision Letter 0]

10 Aug 2022

Response to the Editor-in-Chief and reviewers

We would like to thank both the reviewers for evaluating our study and their constructive criticism, which allowed us to strengthen and clarify our study's conclusions. We have responded in detail to each comment and discussed how the concerns raised were addressed in the revised manuscript.

Journal Requirements

1. Please ensure that your manuscript meets PLOS ONE's style requirements, including those for file naming. The PLOS ONE style templates can be found at https://journals.plos.org/plosone/s/file?id=wjVg/PLOSOne_formatting_sample_main_body.pdf and https://journals.plos.org/plosone/s/file?id=ba62/PLOSOne_formatting_ sample_title_authors_affiliations.pdf 

We revised our manuscript according to The PLOS ONE style templates.

Comments to the Author

3. Have the authors made all data underlying the findings in their manuscript fully available?

We submitted all data sheets (supplemental file) according to your advice.

Reviewer: 1

1. The name of the abscissa axis in Figure 1. B is wrong, and the author wants to express the stimulation time, not the stimulation intensity.

We revised Fig 1.B according to your advice.

2. The special symbols for p-values denoting between-group differences in Figure 2. B and Figure 5. B do not exactly match the legends. 

We modified our legends according to your advice.

3. According to the author's experimental results in Figure 4. A, we found that there are only type IIb muscle fibers in the superficial gastrocnemius muscle in this experiment. How to determine that there are only type IIb muscle fibers in the superficial gastrocnemius muscle?

We confirmed all parts in the ATPase-stained cross-sections, and we examined the deep regions (supplement figure 1, blue square) and the superficial regions (supplement figure 1, red square) separately. From supplement figure 1, the superficial regions consisted entirely of type IIb fibers. In addition, the previous report supported this point1).

1) Kernell D, Lind A, van Diemen AB, De Haan A. Relative degree of stimulation-evoked glycogen degradation in muscle fibres of different type in rat gastrocnemius. J Physiol. 1995;484:139-153. 

4. In the text, the baseline PPT of the gastrocnemius muscles in the LF group was higher than that in the immobilization group. However, in Figure 6. A, the gastrocnemius muscle baseline PPT value in the LF group was lower than in the fixed group.

We retouched the figure of PPT according to your advice.

Reviewer: 2

Major essential changes needed:

1. In this study, myonuclear apoptosis is mentioned as one of the mechanisms of skeletal muscle atrophy, but the involvement of muscle protein synthesis and degradation is ignored. The total and phosphorylated forms of Akt, p70S6K, and 4E-BP1 (FoxO 1/3 phosphorylation would also be informative) in each experimental group should be added as items to be examined. For example, the time course of the acute effect of a single bout of muscle contractile exercise through the belt electrode device on their phosphorylation should be investigated. In particular, Akt-mTOR signaling pathways may orchestrate both a positive protein metabolism in skeletal muscle and myonuclear apoptosis.

Your advice was very important. Thomason DB indicated that muscle protein degradation was more influential than muscle protein synthesis for muscle atrophy in 2-week immobilization1). Atrogin-1 and muscle RING-finger protein (MuRF)-1 as ubiquitin-proteasome pathway enzymes played a main role in muscle protein degradation2). Also, peroxisome proliferator-activated receptor gamma coactivator (PGC)-1α was the main regulator for Atrogin-1 and MuRF-13). Therefore, we examined the mRNA expression of PGC-1α, Atrogin-1, and MuRF-1. As a result, the PGC-1α mRNA expression in the immobilization and LF groups was significantly lower than that in the control group, that in the control and HF groups was no significant difference. Additionally, the Atrogin-1 and MuRF-1 mRNA expressions in the experimental groups were significantly higher than those in the control group, whereas those in the HF group were significantly lower than those in the immobilization and LF groups. From these results, muscle contractile exercise through a belt electrode device may prevent the progression of muscle protein degradation.

These contents were reflected in our manuscript.

1) Thomason DB, Booth FW. Atrophy of the soleus muscle by hindlimb unweighting. J Appl Physiol (1985). 1990;68:1-12.

2) Bodine SC, Latres E, Baumhueter S, Lai VK, Nunez L, Clarke BA et al. Identification of ubiquitin ligases required for skeletal muscle atrophy. J Science. 2001;294:1704-1708.

3) Kang C, Ji LL. Muscle immobilization and remobilization downregulates PGC-1α signaling and the mitochondrial biogenesis pathway. J Appl Physiol 2013;115: 1618-1625.

2. Methods. In statistical analysis for "range of motion on ankle joint dorsiflexion" and "pressure pain threshold", a split-plot ANOVA using 2 factors (time [baseline vs. 1W vs. 2W] and group [control vs. immobilization vs. LF vs. HF]) must be used to determine the interaction and main effects, after verifying the normality of the data.

Thank you for your advice. We modified the statistical process. "range of motion on ankle joint dorsiflexion" and "pressure pain threshold" were assessed using two-way ANOVA, followed by Scheffé’s method. Differences were considered statistically significant at p < 0.05. We added these contents to our manuscript (P10, L218-219).

Minor essential changes needed: 

1. Methods. Please add the length of time between last electrical stimulation, and euthanasia and tissue collection. Exercise itself can raise macrophage accumulation, NGF upregulation, and some cytokines in tissues. This will help exercise physiology readers interpret the results compared to their own.

Based on the previous study, we biopsied 12 hours after the last electrical stimulation1). We added this contents to our manuscript (P8, L169-170).

1) Tanaka M, Morifuji T, Sugimoto K, Akasaka H, Fujimoto T, Yoshikawa M, et al. Effects of combined treatment with blood flow restriction and low-current electrical stimulation on capillary regression in the soleus muscle of diabetic rats. J Appl Physiol (1985). 2021:1219-1229.

2. Methods. The authors described "2 sec (do)/6 sec (rest) duty cycle" as low-frequency and "2 sec (do)/2 sec (rest) duty cycle" as high-frequency for electrical stimulation conditions. It is necessary to clarify the definition of low-frequency or high-frequency in the electrical stimulation conditions.

Dow DE examined the effect of the difference in the number of muscle contractions on muscle atrophy1). Dow DE suggested that 200 contractions per day would be a good design choice to maintain mass, force, and ﬁber CSA while minimizing energy transfer that may negatively affect the tissue and decrease battery life. In our study, the LF and HF groups had 150 and 225 contractions per day. Therefore, we distinguished the 2 groups from the number of muscle contractions.

1) Dow DE, Cederna PS, Hassett CA, Kostrominova TY, Faulkner JA, Dennis RG. Number of contractions to maintain mass and force of a denervated rat muscle. Muscle Nerve. 2004;30:77-86.

3. Methods. To mention that 60% maximal voluntary contraction is most effective in preventing skeletal muscle atrophy, should be provided a clear link to the previous studies referenced. In general, it is defined not only by the intensity of muscle contraction exercise, but also by the number of exercises and the number of sets.

American College of Sports Medicine indicated that the intensity of muscle contractile exercise was one of the important factors for increasing muscle strength1). Concretely, the muscle contractile exercise in 60% MVC produced the largest effect sizes for strength increases1). Furthermore, the same report suggested that the alteration of cross-sectional areas was one of the main components to define muscle strength1). From these, we surmised that the muscle contractile exercise in 60% MVC was most effective to prevent muscle atrophy.

1) American College of Sports Medicine. American College of Sports Medicine position stand. Progression models in resistance training for healthy adults. Med Sci Sports Exerc. 2009;41:687-708.

4. Methods. What is the stimulus intensity you set when you conducted the preliminary experiment to determine the stimulus time?

We set the stimulus intensity to 4.7 mA in the preliminary experiment to determine the stimulus time.

5. Results. For histological observation, the quality of the tissue images stained with immunohistochemistry (Figure 2A) and H&E (Figure 3) were poor in the PDF file.

We modified the dpi of figure 2A and 3.

Minor suggested changes: 

1. Introduction. Hypotheses should be added at the end of the Introductory section.

We revised our manuscript according to your advice.

2. Results. In lines 224 on page 10 to line 226 on page 11, "In the immobilization, LF, and HF groups..." seems to be more appropriate than "In the immobilization, LF, HF groups...".

We modified our manuscript according to your advice (P11, L225).

3. Results. In lines 253-254 on page 12, “The CSA of type IIa myofibers was...” could be more accurate if change to “The CSA of type IIa myofibers in the deep region was...”.

We retouched our manuscript according to your advice (P12, L254 and L257).

---

## [Decision Letter · Decision Letter 1]

12 Sep 2022

Muscle contractile exercise through a belt electrode device prevents myofiber atrophy, muscle contracture, and muscular pain in immobilized rat gastrocnemius muscle

PONE-D-22-09461R1

Dear Dr. Okita,

We’re pleased to inform you that your manuscript has been judged scientifically suitable for publication and will be formally accepted for publication once it meets all outstanding technical requirements.

Kind regards,

Theodoros M. Bampouras

Academic Editor

PLOS ONE

Additional Editor Comments (optional):

Reviewers' comments:

Reviewer's Responses to Questions

**Comments to the Author**

1. If the authors have adequately addressed your comments raised in a previous round of review and you feel that this manuscript is now acceptable for publication, you may indicate that here to bypass the “Comments to the Author” section, enter your conflict of interest statement in the “Confidential to Editor” section, and submit your "Accept" recommendation.

Reviewer #1: All comments have been addressed

Reviewer #2: All comments have been addressed

2. Is the manuscript technically sound, and do the data support the conclusions?

Reviewer #1: Yes

Reviewer #2: Yes

3. Has the statistical analysis been performed appropriately and rigorously? 

Reviewer #1: Yes

Reviewer #2: Yes

4. Have the authors made all data underlying the findings in their manuscript fully available?

Reviewer #1: Yes

Reviewer #2: Yes

5. Is the manuscript presented in an intelligible fashion and written in standard English?

Reviewer #1: Yes

Reviewer #2: Yes

6. Review Comments to the Author

Reviewer #1: The revised version of the manuscript by Honda Y et al. has included satisfactory changes and the queries raised have also been adequately dealt with.

Reviewer #2: The authors have been very responsive to my concerns. I have no more concerns. I agree with their statements and changes in response to the other reviewers as well.

7. PLOS authors have the option to publish the peer review history of their article (what does this mean?). If published, this will include your full peer review and any attached files.

Reviewer #1: **Yes: **Yun Zhou

Reviewer #2: **Yes: **Masahiro Iwata

---

## [Editor Report · Acceptance letter]

16 Sep 2022

PONE-D-22-09461R1 

Muscle contractile exercise through a belt electrode device prevents myofiber atrophy, muscle contracture, and muscular pain in immobilized rat gastrocnemius muscle 

Dear Dr. Okita:

I'm pleased to inform you that your manuscript has been deemed suitable for publication in PLOS ONE. Congratulations! Your manuscript is now with our production department. 

Kind regards, 

on behalf of

Dr. Theodoros M. Bampouras 

Academic Editor

PLOS ONE